# Learning Sample-Specific Models with Low-Rank Personalized Regression

**Benjamin Lengerich**
Carnegie Mellon University
blengeri@cs.cmu.edu

**Bryon Aragam**
University of Chicago
bryon@chicagobooth.edu

**Eric P. Xing**
Carnegie Mellon University
epxing@cs.cmu.edu

## Abstract

Modern applications of machine learning (ML) deal with increasingly heterogeneous datasets comprised of data collected from overlapping latent subpopulations. As a result, traditional models trained over large datasets may fail to recognize highly predictive localized effects in favour of weakly predictive global patterns. This is a problem because localized effects are critical to developing individualized policies and treatment plans in applications ranging from precision medicine to advertising. To address this challenge, we propose to estimate sample-specific models that tailor inference and prediction at the individual level. In contrast to classical ML models that estimate a single, complex model (or only a few complex models), our approach produces a model personalized to each sample. These sample-specific models can be studied to understand subgroup dynamics that go beyond coarse-grained class labels. Crucially, our approach does not assume that relationships between samples (e.g. a similarity network) are known *a priori*. Instead, we use unmodeled covariates to learn a latent distance metric over the samples. We apply this approach to financial, biomedical, and electoral data as well as simulated data and show that sample-specific models provide fine-grained interpretations of complicated phenomena without sacrificing predictive accuracy compared to state-of-the-art models such as deep neural networks.

## 1 Introduction

The scale of modern datasets allows an unprecedented opportunity to infer individual-level effects by borrowing power across large cohorts; however, principled statistical methods for accomplishing this goal are lacking. Standard approaches for adapting to heterogeneity in complex data include random effects models, mixture models, varying coefficients, and hierarchical models. Recent work includes the network lasso [11], the pliable lasso [32], personalized multi-task learning [37], and the localized lasso [38]. Despite this long history, these methods either fail to estimate *individual-level* (i.e. sample-specific) effects, or require prior knowledge regarding the relation between samples (e.g. a network). At the same time, as datasets continue to increase in size and complexity, the possibility of inferring sample-specific phenomena by exploiting patterns in these large datasets has driven interest in important scientific problems such as precision medicine [5, 24]. The relevance and potential impact of sample-specific inference has also been widely acknowledged in applications including psychology [9], education [12], and finance [1].

In this paper, we explore a solution to this dilemma through the framework of "personalized" models. Personalized modeling seeks to estimate a *large* collection of *simple* models in which each model is

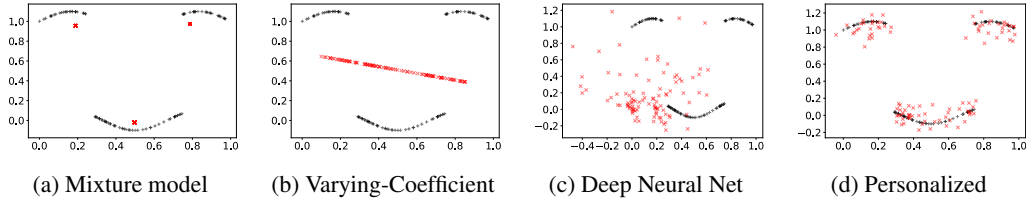

(a) Mixture model    (b) Varying-Coefficient    (c) Deep Neural Net    (d) Personalized

Figure 1: Illustration of the benefits of personalized models. Each point represents the regression parameters for a sample. Black points indicate true effect sizes, while the red points are estimates. Mixture models (a) estimate a limited number of models. The varying-coefficients model (b) estimates sample-specific models but the non-linear structure of the true parameters violates the model assumptions, leading to a poor fit. The locally-linear models induced by a deep learning model (c) do not accurately recover the underlying effect sizes. In contrast, personalized regression (d) accurately recovers effect sizes.

tailored—or "personalized"—to a single sample. This is in contrast to models that seek to estimate a *single*, *complex* model. To make this more precise, suppose we have $n$ samples $(X^{(i)}, Y^{(i)})$, where $Y^{(i)}$ denotes the response and $X^{(i)} \in \mathbb{R}^p$ are predictors. A traditional ML model would model the relationship between $Y^{(i)}$ and $X^{(i)}$ with a single function $f(X^{(i)}; \theta)$ parametrized by a complex parameter $\theta$ (e.g. a deep neural network). In a personalized model, we model each sample with its own function, allowing $\theta$ to be simple while varying with each sample. Thus, the model becomes $Y^{(i)} = f(X^{(i)}; \theta^{(i)})$. These models are estimated jointly with a single objective function, enabling statistical power to be shared between sub-populations.

The flexibility of using different parameter values for different samples enables us to use a simple model class (e.g. logistic regression) to produce models which are simultaneously interpretable and predictive for each individual sample. By treating each sample separately, it is also possible to capture heterogeneous effects *within* similar subgroups (an example of this will be discussed in Section 3.3). Finally, the parameters learned through our framework accurately capture underlying effect sizes, giving users confidence that sample-specific interpretations correspond to real phenomena (Fig 1). Whereas previous work on personalized models either seeks only the population's *distribution* of parameters [34] or requires prior knowledge of the sample relationships [11, 37, 38], we develop a novel framework which estimates sample-specific parameters by adaptively learning relationships from the data. A Python implementation is available at http://www.github.com/blengerich/personalized_regression.

**Motivating Example.**    Consider the problem of understanding election outcomes at the local level. For example, given data on a particular candidate's views and policy proposals, we wish to predict the probability that a particular locality (e.g. county, township, district, etc.) will vote for this candidate. In this example we focus on counties for concreteness. More importantly, in addition to making accurate predictions, we are interested in understanding and explaining how different counties react to different platforms. The latter information—in addition to simple predictive measures—is especially important to candidates and political consultants seeking advantages in major elections such as a presidential election. This information is also important to social and political scientists seeking to understand the characteristics of an electorate and how it is evolving. An application of this motivating example using personalized regression can be found in in Section 3.4.

One approach would be to build individual models for each county, using historical data from previous elections. Immediately we encounter several practical challenges: 1) By building independent models for each county, we fail to share information between related counties, resulting in a loss of statistical power, 2) Since elections are relatively infrequent, the amount of data on each county is limited, resulting in a further loss of power, and 3) To ensure that the models are able to explain the preferences of an electorate, we will be forced to use simple models (e.g. logistic regression or decision trees), which will likely have limited predictive power compared to more complex models. This simultaneous loss of power and predictive accuracy is characteristic of modeling large, heterogeneous datasets arising from aggregating multiple subpopulations. Crucially, in this example the *total* number of samples may be quite large (e.g. there are more than 3,000 US counties and there have been 58 US presidential elections), but the number of samples per subpopulaton is small. Furthermore, these

challenges are in no way unique to this example: similar problems arise for examples in financial, biological, and marketing applications.

One way to alleviate these challenges is to model the $i$th county using a regression model $f(X; \theta^{(i)})$, where the $\theta^{(i)}$ are parameters that vary with each sample and are trained jointly using *all of the data*. This idea of *personalized modeling* allows us to train accurate models using only a single sample from each county—this is useful in settings where collecting more data may be expensive (e.g. biology and medicine) or impossible (e.g. elections and marketing). By allowing the parameter $\theta^{(i)}$ to be sample-specific, there is no longer any need for $f$ to be complex, and simple linear and logistic regression models will suffice, providing useful and interpretable models for each sample.

**Alternative approaches and related work.** One natural approach to heterogeneity is to use mixture models, e.g. a mixture of regression [27] or mixture of experts model [10]. While mixture models present an intriguing way to increase power and borrow strength across the entire cohort, they are notoriously difficult to train and are best at capturing coarse-grained heterogeneity in data. Importantly, mixture models do not capture individual, sample-specific effects and thus cannot model heterogeneity *within* subgroups.

Furthermore, previous approaches to personalized inference [11, 20, 37, 38] assume that there is a *known* network or similarity matrix that encodes how samples in a cohort are related to each other. A crucial distinction between our approach and these approaches is that no such knowledge is assumed. Recent work has also focused on estimating sample-specific parameters for structured models [14, 16, 18, 20, 35]; in these cases, prior knowledge of the graph structure enables efficient testing of sample-specific deviations.

More classical approaches include varying-coefficient (VC) models [8, 13, 30], where the parameter $\theta^{(i)} = \theta(U^{(i)})$ is allowed to depend on additional covariates $U$ in some smooth way, and random effects models [15], where $\theta$ is modeled as a random variable. More recently, the spirit of the VC model has been adapted to use deep neural networks as encoders for complex covariates like images [2, 3] or domain adaptation [26, 29]. In contrast to our approach, which does not impose any regularity or structural assumptions on the model, these approaches typically require strong smoothness (in the case of VC) or distributional (in the case of random effects) assumptions.

Finally, locally-linear models estimated by recent work in model explanations [28] can be interpreted as sample-specific models. We make explicit comparisons to this approach in our experiments (Section 3), but we point out here that local explanations serve to interpret a black-box model—which may be incorrect—and not the true mechanisms underlying the data. This is clearly illustrated in Fig 1c, where local linear approximations do a good job of explaining the behaviour of the underlying neural network, but nonetheless fail to capture the true regression coefficients. This tradeoff between inference and prediction is well-established in the literature.

## 2 Learning sample-specific models

For clarity, we describe the main idea using a linear model for each personalized model; extension to arbitrary generalized linear models including logistic regression is straightforward. In Section 3, we include experiments using both linear and logistic regression. A traditional linear model would dictate $Y^{(i)} = \langle X^{(i)}, \theta \rangle + w^{(i)}$, where the $w^{(i)}$ are noise and the parameter $\theta \in \mathbb{R}^p$ is shared across different samples. We relax this model by allowing $\theta$ to vary with each sample, i.e.

$$Y^{(i)} = \langle X^{(i)}, \theta^{(i)} \rangle + w^{(i)}. \tag{1}$$

Clearly, without additional constraints, this model is overparametrized—there is a $(p-1)$-dimensional subspace of solutions to the equation $Y^{(i)} = \langle X^{(i)}, \theta^{(i)} \rangle$ in $\theta^{(i)}$ for each $i$. Thus, the key is to choose a solution $\theta^{(i)}$ that simultaneously leads to good generalization and accurate inferences about the $i$th sample. We propose two strategies for this: (a) a low-rank latent representation of the parameters $\theta^{(i)}$ and (b) a novel regularization scheme.

### 2.1 Low-rank representation

We constrain the matrix of personalized parameters $\Omega = [\theta^{(1)} \mid \cdots \mid \theta^{(n)}] \in \mathbb{R}^{p \times n}$ to be low-rank, i.e. $\theta^{(i)} = Q^T Z^{(i)}$ for some loadings $Z^{(i)} \in \mathbb{R}^q$ and some dictionary $Q \in \mathbb{R}^{q \times p}$. Letting $Z \in \mathbb{R}^{q \times n}$

denote the matrix of loadings, we have a low-rank representation of $\Omega = Q^T Z$. The choice of $q$ is determined by the user's desired latent dimensionality; for $q \ll p$, using only $\Theta\big(q(n+p)\big)$ instead of the $\Theta(np)$ of a full-rank solution can greatly improve computational and statistical efficiency. In addition, the low-rank formulation enables us to use $\ell_2$ distance in $Z$ in Eq. (4) to restrict Euclidean distances between the $\theta^{(i)}$: After normalizing the columns of $Q$, we have

$$\|\theta^{(i)} - \theta^{(j)}\| \leq \sqrt{p}\|Z^{(i)} - Z^{(j)}\|. \tag{2}$$

This illustrates that closeness in the loadings $Z^{(i)}$ implies closeness in parameters $\theta^{(i)}$. This fact will be exploited to regularize $\theta^{(i)}$ (Section 2.2).

This use of a dictionary $Q$ is common in multi-task learning [23] based on the assumption that tasks inherently use shared atomic representations. Here, we make the analogous assumption that samples arise from combinations of shared processes, so sample-specific models based on a shared dictionary efficiently characterize sample heterogeneity. Sparsity in $\theta$ can be realized by sparsity in $Z, Q$; for instance, effect sizes which are consistently zero across all samples can be created by zero vectors in the columns of $Q$. The low-rank formulation also implicitly constrains the number of personalized sparsity patterns; this can be adjusted by changing the latent dimensionality $q$.

## 2.2 Distance-matching

Existing approaches [11, 20, 37, 38] assume that there is a known weighted network $(\lambda_{ij})_{i,j=1}^n$ over samples such that $\|\theta^{(i)} - \theta^{(j)}\| \approx \lambda_{ij}$. In other words, we have prior knowledge of which parameters should be similar. We avoid this strong assumption by instead assuming that we have additional covariates $U^{(i)} \in \mathbb{R}^k$ for which *there exists* some way to measure similarity that corresponds to similarity in the parameter space, however, we don't have advance knowledge of this. More specifically, we regularize the parameters $\theta^{(i)}$ by requiring that similarity in $\theta$ corresponds to similarity in $U$, i.e. $\|\theta^{(i)} - \theta^{(j)}\| \approx \rho(U^{(i)}, U^{(j)})$, where $\rho$ is an *unknown*, latent metric on the covariates $U$. In applications, the $U^{(i)}$ represent exogenous variables that we do not wish to directly model; for example, in our motivating example of an electoral analysis, this may include demographic information about the localities.

To promote similar structures in parameters as in covariates, we adapt a distance-matching regularization (DMR) scheme [17] to penalize the squared difference in implied distances. The covariate distances are modeled as a weighted sum:

$$\rho_\phi(u, v) = \sum_{\ell=1}^k \phi_\ell d_\ell(u_\ell, v_\ell), \quad \phi_\ell \geq 0, \tag{3}$$

where each $d_\ell$ ($\ell = 1, \ldots, k$) is a metric for a covariate. The positive vector $\phi$ represents a linear transformation of these "simple" distances into more useful latent distance functions. By using a linear parametrization for $\rho_\phi$, we can interpret the learned effects by inspecting the weights assigned to each covariate.

By Eq. (2), in order for $\|\theta^{(i)} - \theta^{(j)}\| \approx \rho_\phi(U^{(i)}, U^{(j)})$, it suffices to require $\|Z^{(i)} - Z^{(j)}\| \approx \rho_\phi(U^{(i)}, U^{(j)})$. With this in mind, define the following *distance-matching regularizer*:

$$D_\gamma^{(i)}(Z, \phi) = \frac{\gamma}{2} \sum_{j \in B_r(i)} \big(\rho_\phi(U^{(i)}, U^{(j)}) - \|Z^{(i)} - Z^{(j)}\|^2\big)^2, \tag{4}$$

where $B_r(i) = \{j : \|Z^{(i)} - Z^{(j)}\|^2 < r\}$. This regularizer promotes imitating the structure of covariate values in the regression parameters. By using $Z$ instead of $\Omega$ in the regularizer, calculation of distances is much more efficient when $q \ll p$. A discussion of hyperparameter selection is contained in Section. B.3 of the supplement.

## 2.3 Personalized Regression

Let $\ell(x, y, \theta)$ be a loss function, e.g. least-squares or logistic loss. For each sample $i$ of the training data, define a regularized, sample-specific loss by

$$\mathcal{L}^{(i)}(Z, Q, \phi) = \ell(X^{(i)}, Y^{(i)}, Q^T Z^{(i)}) + \psi_\lambda(Q^T Z^{(i)}) + D_\gamma^{(i)}(Z, \phi), \tag{5}$$

where $\psi_\lambda$ is a regularizer such as the $\ell_1$ penalty and $D_\gamma^{(i)}$ is the distance-matching regularizer defined in Eq. (4). We learn $\Omega$ and $\phi$ by minimizing the following composite objective:

$$\mathcal{L}(Z, Q, \phi) = \sum_{i=1}^{n} \mathcal{L}^{(i)}(Z, Q, \phi) + \upsilon\|\phi - 1\|_2^2, \tag{6}$$

where the second term regularizes the distance function $\rho_\phi$ with strength set by $\upsilon$, and we recall that $\Omega = Q^T Z$. The hyperparameter $\gamma$ trades off sensitivity to prediction of the response variable against sensitivity to covariate structure.

**Optimization.** We minimize the composite objective $\mathcal{L}(Z, Q, \phi)$ with subgradient descent combined with a specific initialization and learning rate schedule. An outline of the algorithm can be found in Alg. 1 below. In detail, we initialize $\Omega$ by setting $\theta^{(i)} \sim N(\widehat{\theta}^{\text{pop}}, \epsilon I)$ for a population model $\widehat{\theta}^{\text{pop}}$ such as the Lasso or elastic net and then initialize $Z$ and $Q$ by factorizing $\Omega$ with PCA. $\epsilon$ is a very small value used only to enable factorization by the PCA algorithm. Each personalized estimator is endowed with a personalized learning rate $\alpha_t^{(i)} = \alpha_t / \|\widehat{\theta}_t^{(i)} - \widehat{\theta}^{(\text{pop})}\|_\infty$, which scales the global learning rate $\alpha_t$ according to how far the estimator has traveled. In addition to working well in practice, this scheme guarantees that the center of mass of the personalized regression coefficients does not deviate too far from the intialization $\widehat{\theta}^{\text{pop}}$, even though the coefficients $\widehat{\theta}^{(i)}$ remain unconstrained. This property is discussed in more detail in Section 2.4.

---

**Algorithm 1** Personalized Estimation

---

**Require:** $\widehat{\theta}^{pop}, \lambda, \gamma, \upsilon, \alpha, c$
1: $\theta^{(1)}, \ldots, \theta^{(n)} \leftarrow \widehat{\theta}^{pop}$
2: $\Omega \leftarrow [\theta^{(1)} | \ldots | \theta^{(n)}]$
3: $Z, Q \leftarrow \text{PCA}(\Omega)$
4: $\phi \leftarrow \mathbf{1}$
5: $\alpha \leftarrow \alpha_0$
6: **do**
7: $\quad \widetilde{Z}, \widetilde{Q}, \widetilde{\phi} \leftarrow Z, Q, \phi$
8: $\quad \phi \leftarrow \phi - \alpha\frac{\partial}{\partial\phi}\mathcal{L}(\widetilde{Z}, \widetilde{Q}, \widetilde{\phi}; \lambda, \gamma, \upsilon)$
9: $\quad Z^{(i)} \leftarrow Z^{(i)} - \frac{\alpha}{\|\theta^{(i)} - \widehat{\theta}^{\text{pop}}\|_\infty}\big[\frac{\partial}{\partial Z^{(i)}} \sum_{i=1}^{n} D_\gamma^{(i)}(\widetilde{Z}, \widetilde{\phi}) +$
$\qquad \widetilde{Q}\big(\partial\ell(X^{(i)}, Y^{(i)}, \theta^{(i)}) + \partial\psi_\lambda(\theta^{(i)}))\big] \quad \forall\, i \in [1, \ldots, n]$
10: $\quad Q \leftarrow Q - \alpha\big[\frac{\partial}{\partial Q} \sum_{i=1}^{n} D_\gamma^{(i)}(\widetilde{Z}, \widetilde{\phi}) + \sum_{i=1}^{n} \widetilde{Z}^{(i)}\big(\partial\ell(X^{(i)}, Y^{(i)}, \theta^{(i)})^T + \partial\psi_\lambda(\theta^{(i)})^T\big)\big]$
11: $\quad \alpha \leftarrow \alpha c$
12: $\quad \theta^{(i)} \leftarrow Q^T Z^{(i)} \quad \forall\, i \in [1, \ldots, n]$
13: $\quad \Omega \leftarrow [\theta^{(1)} | \ldots | \theta^{(n)}]$
14: **while** not converged
15: **return** $\Omega, Z, Q, \phi$

---

**Prediction.** Given a test point $(X, U)$, we form a sample-specific model by averaging the model parameters of the $k_n$ nearest training points, according to the learned distance metric $\rho_\phi$:

$$\theta = \frac{1}{k_n} \sum_{j=1}^{k_n} \theta^{(\eta(\rho_\phi, U)[j])}, \qquad \eta(\rho_\phi, U) = \underset{1 \leq i \leq n}{\text{argsort}} \, \rho_\phi(U, U^{(i)}). \tag{7a}$$

Increasing $k_n$ drives the test models toward the population model to control overfitting. In our experiments, we use $k_n = 3$.

We have intentionally avoided using $X$ to select $\theta$ so that interpretation of $\theta$ is not confounded by $X$. In some cases, however, the sample predictors can provide additional insight to sample distances (e.g. [36]); we leave it to future work to examine how to augment estimations of sample distances by including distances between predictors.

**Scalability.** Naïvely, the distance-matching regularizer has $O(n^2)$ pairwise distances to calculate, however this calculation can be made efficient as follows. First, the terms involving $d_\ell(U_\ell^{(i)}, U_\ell^{(j)})$ remain unchanged during optimization, so that their computation can be amortized. This allows the use of feature-wise distance metrics which are computationally intensive (e.g. the output of a deep learning model for image covariates). Furthermore, these values are never optimized, so the distance metrics $d_\ell$ need not be differentiable. This allows for a wide variety of distance metrics, such as the discrete metric for unordered categorical covariates. Second, we streamline the calculation of nearest neighbors in two ways: 1) Storing $Z$ in a spatial data structure and 2) Shrinking the hyperparameter $r$ used in (4). With these performance improvements, we are able to fit models to datasets with over 10,000 samples and 1000s of predictors on a Macbook Pro with 16GB RAM in under an hour.

## 2.4 Analysis

Initializing sample-specific models around a population estimate is convenient because the sample-specific estimates do not diverge from the population estimate unless they have strong reason to do so. Here, we analyze linear regression minimized by squared loss (e.g., $f(X^{(i)}, Y^{(i)}, \theta^{(i)}) = (Y^{(i)} - X^{(i)}\theta^{(i)})^2$), though the properties extend to any predictive loss function with a Lipschitz-continuous subgradient.

**Theorem 1.** *Let us consider personalized linear regression with $\psi_\lambda(x) = \lambda\|x\|_1$ (i.e. $\ell_1$ regularization). Let $X$ be normalized such that $\max_i \|X^{(i)}\|_\infty \leq 1$, $\|X^{(i)}\|_1 = 1$.*

*Define $\overline{\theta}_t := \frac{1}{n}\sum_{i=1}^n \widehat{\theta}_t^{(i)}$, where $\widehat{\theta}_t^{(i)}$ is the current value of $\widehat{\theta}^{(i)}$ after $t$ iterations.*

*Let the overall learning rate follow a multiplicative decay such that $\alpha_t = \alpha_0 c^t$, where $\alpha_0$ is the initial learning rate and $c$ is a constant decay factor. Then at iteration $\tau$,*

$$\|\overline{\theta}_\tau - \widehat{\theta}^{\text{pop}}\|_\infty \in \mathcal{O}(\lambda). \tag{8}$$

That is, the center of mass of the personalized regression coefficients does not deviate too far from the initialization $\widehat{\theta}^{\text{pop}}$, even though the coefficients $\widehat{\theta}^{(i)}$ remain unconstrained. In addition, the distance-matching regularizer does not move the center of mass and the update to the center of mass does not grow with the number of samples. Proofs of these claims are included in Appendix A of the supplement.

## 3 Experiments

We compare personalized regression (hereafter, PR) to four baselines: 1) Population linear or logistic regression, 2) A mixture regression (MR) model, 3) Varying coefficients (VC), 4) Deep neural networks (DNN). First, we evaluate each method's ability to recover the true parameters from simulated data. Then we present three real data case studies, each progressively more challenging than the previous: 1) Stock prediction using financial data, 2) Cancer diagnosis from mass spectrometry data, and 3) Electoral prediction using historical election data. The results are summarized in Table 1 for easy reference. Details on all the algorithms and datasets used, as well as additional results and figures, can be found in Appendix B of the supplement.

We believe the out-of-sample prediction results provide strong evidence that any harmful overfitting of PR is outweighed by the benefit of personalized estimation. This agrees with famous results such as [31], where it is showed that optimal ensembles of linear models consist of overfitted atoms; see especially Eq. 12 and Fig. 2 therein.

## 3.1 Simulation Study

We first investigate the capacity of personalized regression to recover true effect sizes in a small-scale simulation study. We generate $X_j \sim \text{Unif}(-1, 1)$ ($j = 1, 2$), $U \sim \text{Unif}(0, 1)$, $\theta^{(i)} = [U^{(i)}, \mathcal{I}_{|U^{(i)}|>0.5} + 0.1\sin(U^{(i)})] \in \mathbb{R}^2$, and $Y^{(i)} = X^{(i)}\theta^{(i)} + w^{(i)}$, with $w^{(i)} \sim N(0, 0.1)$. As shown in Fig. 1, this produces regression parameters with a discontinuous distribution. The algorithms are given both $X$ and $U$ as input during training, and we use LIME [28] to generate local linear approximations to the DNN in order to estimate parameters $\theta^{(i)}$ for each sample. In this setting, there

Table 1: Predictive performance on test sets. For continuous response variables, we report correlation coefficient ($R^2$) and mean squared error (MSE) of the predictions. For classification tasks, we report area under the receiver operating characteristic curve (AUROC) and the accuracy (ACC). For the simulation, we also report recovery error of the true regression parameters in the training set, with (mean $\pm$ std) values calculated over 20 experiments with different values of $X, U, w$.

| Model | Simulation | | | Financial | | Cancer | | Election | |
|---|---|---|---|---|---|---|---|---|---|
| | $\|\widehat{\Omega} - \Omega\|_2$ | $R^2$ | MSE | $R^2$ | MSE | AUROC | Acc | $R^2$ | MSE |
| Pop. | $24.76 \pm 0.02$ | $0.57 \pm 0.03$ | $0.133 \pm 0.01$ | 0.01 | 64144 | 0.794 | 0.962 | 0.00 | 0.019 |
| MR | $19.31 \pm 0.87$ | $0.83 \pm 0.03$ | $0.054 \pm 0.01$ | 0.74 | 16146 | 0.876 | 0.939 | $-0.56$ | 0.031 |
| VC | $24.88 \pm 0.09$ | $0.66 \pm 0.02$ | $0.106 \pm 0.01$ | 0.06 | 60694 | 0.430 | 0.863 | 0.00 | 0.019 |
| DNN | $30.29 \pm 0.55$ | $0.91 \pm 0.03$ | $0.028 \pm 0.01$ | $-0.02$ | 63028 | 0.901 | 0.955 | 0.00 | 0.019 |
| PR | $\mathbf{9.02 \pm 2.53}$ | $\mathbf{0.936 \pm 0.05}$ | $\mathbf{0.020 \pm 0.01}$ | **0.86** | **4822** | **0.923** | **0.975** | **0.45** | **0.011** |

exists a discontinuous function which could output exactly the sample-specific regression models from the covariates that a neural network should be able to learn accurately. In this sense, the neural network is "correctly specified" for this dataset, testing how well locally-linear models approximate the true parameters. More extensive simulation experiments, with varying $n$ and $p$ are available in Sec. C.1 of the Supplement.

**Results.** The results are presented in Table 1 and visualized in Fig. 1. As expected, the recovery error is much lower for PR, while the DNN shows competitive predictive error. The population estimator successfully recovers the mean effect sizes, but this central model is not accurate for any individual, resulting in poor performance both in recovering $\Omega$ and in prediction. Similarly, both MR and VC perform poorly. As expected, the deep learning model excels at predictive error, however, the local linear approximations do not accurately recover the sample-specific linear models. In contrast, PR exhibits both the flexibility and the structure to learn the true regression parameters while retaining predictive performance.

## 3.2 Financial Prediction

A common task in financial trading is to predict the price of a security at some point in the future. This is a challenging task made more difficult by nonstationarity—the interpretation of an event changes over time, and different securities may respond to the same event differently. We built a dataset of security prices over a 30-year time frame by joining stock and ETF trading histories to a database of global news headlines (details in supplement). The predictors $X^{(i,t)}$ consist of the trading history of the 24 securities over the previous 2 weeks as well as global news headlines from the same time period. The covariates $U^{(i,t)}$ consist of the date and security characteristics (name, region, and industry). The target $Y^{(i,t)}$ is the price of this security 2 weeks after $t$.

**Results.** PR significantly outperforms baseline methods to predict price movements (Table 1). In contrast to standard models which average effects over long time periods and/or securities, PR summarizes gradual shifts in attention. The estimated sample-specific models are visualized in Fig. 2. The strongest clustering behavior is due to time (Fig. 2b). For instance, models fit to samples in the era of U.S. "stagflation" (1973-1975) are overlaid on models for samples in the early 1990s U.S. recession. In both of these cases, real equity prices declined against the background of high inflation rates. In contrast, the recessions marked by structural problems such as the Great Financial Crisis of 2008 are separated from the others. Within each time period, we also see that industries (Fig. S4a), regions (Fig. S4b), and securities (Fig. S4c) are strongly clustered (details in supplement).

## 3.3 Cancer Analysis

In cancer analysis, the challenges of sample heterogeneity are paramount and well-known. Increasing biomedical evidence suggests that patients do not fall into discrete clusters [5, 21], but rather each patient experiences a unique disease that should be approached from an individualized perspective [7]. Here, we investigate the capacity of PR to distinguish malignant from benign skin lesions using a dataset of desorption electrospray ionization mass spectrometry imaging (DESI-MSI) of a common skin cancer, basal cell carcinoma (BCC) [22] (details in supplement).

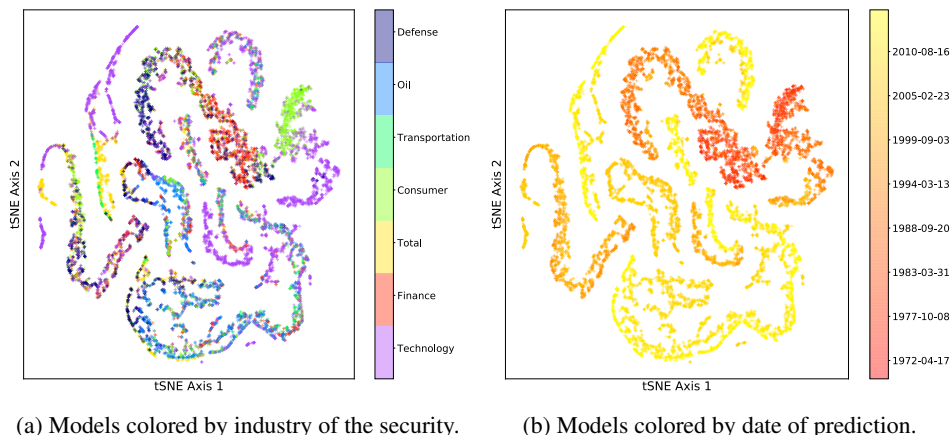

(a) Models colored by industry of the security.    (b) Models colored by date of prediction.

Figure 2: Personalized financial models using t-SNE [33] embedding. Each point represents a regression model for one security at a single date.

**Results.**    As shown in Table 1, PR produces the best predictions of tumor status amongst the methods evaluated. The substantial improvement over competing methods is likely due to the long tail of the distribution of characteristic features—we observe that the number of samples which assign the largest influence to each feature has a long tail (Fig. S7). By summing the most important features for each instance, we can transform these sample-specific explanations into patient-specific explanations (Table S4). These explanations depict a clustering of patients in which there are 8 distinct subtypes (visualized in Fig. S6). While we may hope that a mixture model could recover these patient clusters, actual mixture components are less accurate in prediction (Table 1), likely due to their independent estimation and reduced statistical power. Furthermore, this clustering by patient is incomplete—there is also significant heterogeneity in the models for each patient (Fig. S6). This may point to the "mosaic" view of tumors, under which single tumors are comprised of multiple cell lines [19]. This example underscores the benefits of treating sample heterogeneity as fundamental by designing algorithms to estimate sample-specific models.

### 3.4 Presidential Election Analysis

Our last experiment illustrates a practical use case for the example of modeling election outcomes discussed in Section 1. The goals are twofold: 1) To predict county-level election results, and 2) To explore the use of distinct regression models as embeddings of samples in order to better understand voting preferences at the county (i.e. sample-specific) level. The data are from the 2012 U.S. presidential election and consist of discrete representations of each candidate based on candidate positions while the outcomes are the county-level vote proportions (details in supplement). For the covariates $U$, we used county demographic information from the 2010 U.S. Census. As the outcome varies across samples but the predictors remain constant, the personalized regression models must encode sample heterogeneity by estimating different regression parameters for different samples, thus creating county representations ("embeddings") which combine both voting and demographic data.

**Results.**    The out-of-sample predictive error is significantly reduced by personalization (Table 1). Figs. 3, S8 depict embeddings of the Pennsylvania counties included in the training set. Generating county embeddings based solely on voting outcome constrains the embeddings near a one-dimensional manifold (Fig. S8b), while demographics produce embeddings which do not strongly correspond to voting patterns (Fig. 3a). In contrast, the personalized models produce a structure which interpolates between the two types of data (Fig. 3b). An interesting case is that of the Lackawanna and Allegheny counties. While these counties had similar voting results in the 2012 election, their embeddings are far apart due to the difference in demographics between their major metropolitan areas. This indicates that the county populations may be voting for different reasons despite similar demographics, a finding that is not discovered by jointly inspecting the demographic and voting data (Fig. S8e). Thus, sample-specific models can be used to understand the complexities of election results.

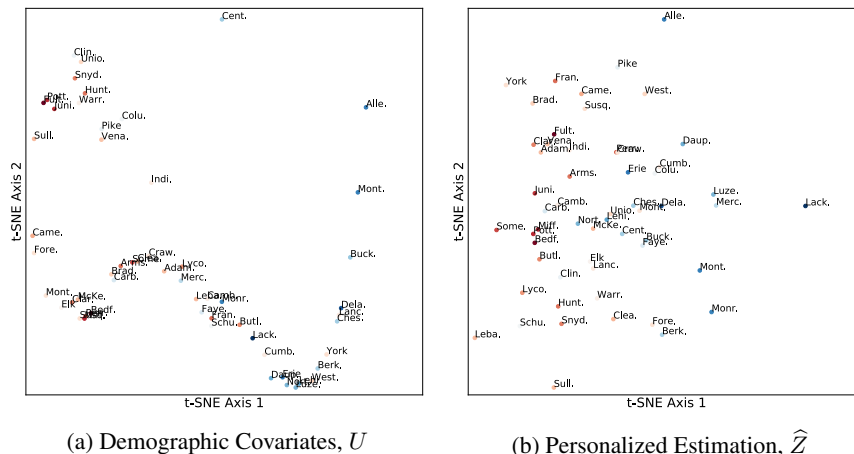

(a) Demographic Covariates, $U$

(b) Personalized Estimation, $\widehat{Z}$

Figure 3: Embeddings of Pennsylvania counties. Each point represents a county, with color gradient corresponding to the 2012 election result (red for Republican candidate, blue for Democratic candidate). Due to space constraints, the name of each county has been abbreviated, with a key in Table S5 of the Supplement. (a) The raw covariates $U$ lie near a low-dimensional manifold that does not correspond to voter outcome. (b) Personalized regression models form embeddings ($\widehat{Z}$) which interpolate between demographic and voting information.

## 4   Discussion and Future Work

We have presented a framework to estimate collections of models by matching structure in sample covariates to structure in regression parameters. We showed that this framework accurately recovers sample-specific parameters, enabling collections of simple models to surpass the predictive capacity of larger, uninterpretable models. Our framework also enables fine-grained analyses which can be used to understand sample heterogeneity, even within groups of similar samples. Beyond estimating sample-specific models, we also believe it would be possible to adapt these ideas to improve standard models. For instance, the distance-matching regularizer may be applied to augment standard mixture models. It would also be interesting to consider extensions of this framework to more structured models such as personalized probabilistic graphical models. Overall, the success of these personalized models underscores the importance of directly treating sample heterogeneity rather than building increasingly-complicated cohort-level models.

#### Acknowledgments

We thank Maruan Al-Shedivat, Gregory Cooper, and Rich Caruana for insightful discussion.

This material is based upon work supported by NIH R01GM114311. Any opinions, findings and conclusions or recommendations expressed in this material are those of the author(s) and do not necessarily reflect the views of the National Institutes of Health.

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
