[Supplementary Material]

# A Analysis

Initializing sample-specific models around a population estimate is convenient because the sample-specific estimations do not diverge from the population estimate unless they have strong reason to do so. Here, we analyze linear regression minimized by squared loss (e.g., $f(X^{(i)}, Y^{(i)}, \theta^{(i)}) = (Y^{(i)} - X^{(i)}\theta^{(i)})^2$), though the properties extend to any predictive loss function with a Lipschitz-continuous subgradient.

We prove Theorem 1 by introducing two helpful lemmas. First, the distance-matching regularizer does not move the center of mass:

**Lemma 2.** *For any $Z, \phi$*

$$\sum_{i=1}^{n} \sum_{j \in B_r(i)} \frac{\partial}{\partial \theta^{(i)}} D_\gamma^{(j)}(Z, \phi) = \mathbf{0}.$$

*Proof of Lemma 2.* Let $g(i,j) = \mathcal{I}_{\{j \in B_r(i)\}} \frac{\partial}{\partial \theta^{(i)}} \left( \rho_\phi(U^{(i)}, U^{(j)}) - \|Z^{(i)} - Z^{(j)}\|^2 \right)^2$. Then, for any symmetric $\rho_\phi$, $\forall\, i, j \in \{1, ..., n\}$,

$$g(i,j) = 2\mathcal{I}_{\{j \in B_r(i)\}} \left( \rho_\phi(U^{(i)}, U^{(j)}) - \|Z^{(i)} - Z^{(j)}\|^2 \right)(-2)(Z^{(i)} - Z^{(j)}) \tag{9a}$$

$$= -2\mathcal{I}_{\{i \in B_r(j)\}} \left( \rho_\phi(U^{(i)}, U^{(j)}) - \|Z^{(i)} - Z^{(j)}\|^2 \right)(-2)(Z^{(j)} - Z^{(i)}) \tag{9b}$$

$$= -g(j,i) \tag{9c}$$

. So

$$\sum_{i=1}^{n} \sum_{j \in B_r(i)} \frac{\partial}{\partial \theta^{(i)}} D_\gamma^{(j)}(Z, \phi) = \sum_{i=1}^{n} \sum_{j=1}^{n} g(i,j) \tag{9d}$$

$$= \sum_{i=1}^{n} g(i,i) = \mathbf{0} \qquad \square$$

This implies that the distance-matching regularizer has no effect on $\bar{\theta}$. An intuitive explanation is to visualize each $D_\gamma^{(i)}(Z, \phi)$ as a collection of springs connecting estimator $i$ to each of the other estimators. While the springs will have some control over the pairwise distances, they cannot move the center of mass of any pair of particles and thus cannot adjust the center of mass of the system.

Second, the update to the center of mass does not grow with the number of samples:

**Lemma 3.** *At iteration $t$, the update to the center of mass is bounded by:*

$$\|\bar{\theta}_{t+1} - \bar{\theta}_t\|_\infty \leq \alpha_t(\lambda + 1) \tag{10}$$

*Proof of Lemma 3.* The update to the barycenter at iteration $t$ is:

$$\bar{\theta}_{t+1} - \bar{\theta}_t = \frac{1}{n} \Big[ \sum_{i=1}^{n} \alpha_t^{(i)} \frac{\partial}{\partial \theta^{(i)}} \ell(X^{(i)}, Y^{(i)}, \theta^{(i)}) + \alpha_t^{(i)} \psi'(\theta^{(i)}) \tag{11a}$$

$$+ \sum_{i=1}^{n} \alpha_t^{(i)} \sum_{j \in B_r(i)} \frac{\partial}{\partial \theta^{(i)}} D_\gamma^{(j)}(Z, \phi) \Big] \tag{11b}$$

$$= \frac{1}{n} \Big[ \sum_{i=1}^{n} \alpha_t^{(i)} \frac{\partial}{\partial \theta^{(i)}} \ell(X^{(i)}, Y^{(i)}, \theta^{(i)}) + \alpha_t^{(i)} \psi'(\theta^{(i)}) \Big] \qquad \text{by Lemma 2} \tag{11c}$$

where $\psi'(\cdot)$ is the sub-gradient of $\psi_\lambda(\cdot)$ used for optimization. Let $\ell^{(i)}(\cdot) = \ell'(Y^{(i)}, X^{(i)}, \cdot)$ where $\ell'$ is the subgradient of $\ell$ used for optimization, with $\ell^{(i)}$ $k^{(i)}$-Lipschitz continuous. Let $\psi'$ be $k_\psi$

Lipschitz-continuous. Then

$$\|\bar{\theta}_{t+1} - \bar{\theta}_t\|_\infty = \frac{1}{n}\|\sum_{i=1}^{n} \alpha_t^{(i)}\big(\ell^{(i)}(\widehat{\theta}_t^{(i)}) + \psi_\lambda'(\widehat{\theta}_t^{(i)})\big)\|_\infty \tag{11d}$$

$$\leq \frac{1}{n}\|\sum_{i=1}^{n} \alpha_t^{(i)}\big(k^{(i)}|\widehat{\theta}_t^{(i)} - \widehat{\theta}^{\mathrm{pop}}| + \ell^{(i)}(\widehat{\theta}^{\mathrm{pop}}) + \psi_\lambda'(\widehat{\theta}^{\mathrm{pop}}) + \psi_\lambda'(\widehat{\theta}_t^{(i)}) - \psi_\lambda'(\widehat{\theta}^{\mathrm{pop}})\big)\|_\infty \tag{11e}$$

$$= \frac{1}{n}\|\sum_{i=1}^{n} \alpha_t^{(i)}\big(k^{(i)}|\widehat{\theta}_t^{(i)} - \widehat{\theta}^{\mathrm{pop}}| + \psi_\lambda'(\widehat{\theta}_t^{(i)}) - \psi_\lambda'(\widehat{\theta}^{\mathrm{pop}})\big)\|_\infty \tag{11f}$$

$$\leq \frac{1}{n}\|\sum_{i=1}^{n} \alpha_t^{(i)}\big(k^{(i)}|\widehat{\theta}_t^{(i)} - \widehat{\theta}^{\mathrm{pop}}| + k_\psi|\widehat{\theta}_t^{(i)} - \widehat{\theta}^{\mathrm{pop}}|\big)\|_\infty \tag{11g}$$

$$\leq (\max_i k^{(i)} + k_\psi)\max_i \alpha_t^{(i)}\|\widehat{\theta}_t^{(i)} - \widehat{\theta}^{\mathrm{pop}}\|_\infty \tag{11h}$$

where 11f holds because the population estimator $\widehat{\theta}^{\mathrm{pop}}$ was solved by a gradient descent algorithm resulting in $\sum_{i=1}^{n} \ell^{(i)}(\widehat{\theta}^{\mathrm{pop}}) = -n\psi'(\widehat{\theta}^{\mathrm{pop}})$. For $\psi'(x) = \begin{cases} -\lambda & x < 0 \\ \lambda & x > 0 \\ 0 & x = 0 \end{cases}$, as used in our experiments, $k_\psi = \lambda$. For linear regression, $k^{(i)} = \|X^{(i)}\|_1 \max_j X_j^{(i)}$. For $X$ normalized so that $\|X^{(i)}\|_1 = 1$ and $\|X^{(i)}\|_\infty \leq 1$, $\max_i k^{(i)} \leq 1$. With $\alpha_t^{(i)} = \frac{\alpha_t}{\|\widehat{\theta}_t^{(i)} - \widehat{\theta}^{\mathrm{pop}}\|_\infty}$, we have the upper bound on the update:

$$\|\bar{\theta}_{t+1} - \bar{\theta}_t\|_\infty \leq \alpha_t(\lambda + 1). \qquad \square$$

Now we are ready to prove Theorem 1.

*Proof of Theorem 1.* For a multiplicative decay on the learning rate $\alpha_t \leq \alpha_0 c^t$,

$$\|\bar{\theta}_\tau - \widehat{\theta}^{\mathrm{pop}}\|_\infty \leq \sum_{t=1}^{\tau}\|\bar{\theta}_t - \bar{\theta}_{t-1}\|_\infty$$

$$\leq \sum_{t=1}^{\tau} \alpha_t(\lambda + 1) \qquad \text{by Lemma 3}$$

$$\leq \alpha_0(\lambda + 1)\sum_{t=1}^{\tau} c^t$$

$$\leq \alpha_0(\lambda + 1)\frac{1 - c^\tau}{1 - c}. \qquad \square$$

# B  Experiment details

## B.1  Baselines

For each experiment, we use several baseline models to benchmark performance:

- *Population model.* First, we use elastic net regularization [40] as a generalizable population estimator.

- *Mixture of regressions.* To estimate a small collection of models, we use a standard mixture model optimized by expectation-maximization. Since this model does not share information between mixture components, the number of components must be much smaller than the number of samples.

- *Varying coefficient model.* To estimate sample-specific models, we use an $\ell_1$-regularized linear varying-coefficients model [13].

- *Deep neural network.* Finally, to compare against models with large representational capacity, we include a neural network. This neural network contains 5 hidden layers, with layer sizes and nonlinearities treated as hyperparameters optimized for cross-validation loss by grid search. The final version contains 250 hidden nodes in each layer with sigmoid nonlinearities.

For the tasks with continuous outcomes, these are linear regression models; for classification tasks, these are logistic regression models.

## B.2 Subgradients

For personalized linear regression with $\ell_1$ regularization, we use the standard subgradients:

- $\ell'(Y^{(i)}, X^{(i)}, \theta^{(i)}) = -2(Y^{(i)} - X^{(i)}\theta^{(i)})X^{(i)}$

- $\psi'(x) = \begin{cases} -\lambda & x < 0 \\ \lambda & x > 0 \\ 0 & x = 0 \end{cases}$

. For personalized logistic regression, we use $\ell'(Y^{(i)}, X^{(i)}, \theta^{(i)}) = X^{(i)}\left(\frac{\exp{(X^{(i)T}\theta^{(i)})}}{1+\exp{(X^{(i)T}\theta^{(i)})}} - Y^{(i)}\right)$.

## B.3 Hyperparameter selection

While the personalized regression approaches estimates a large number of parameters, there are relatively few hyperparameters. Hyperparameters to be selected are: $\lambda$ the strength of the traditional regression regularizer, $\gamma$ the strength of the distance-matching regularizer, $r$ the diameter of the neighborhoods considered by the distance-matching regularizer, $\upsilon$ the strength of regularizer on $\phi$, and $q$ the latent dimensionality. $\lambda$ should be set equivalent to the $\lambda$ used in the population estimator. $\gamma$ requires some tuning and should be set such that the distance-matching regularizer contributes the a same order of magnitude on the total loss as does the predictive loss. $r$ should be set to reflect the user's desired neighborhood of personalization; larger $r$ produces personalized estimates which reflect covariate distances even for very different samples, smaller $r$ improves computation speed but decreases the size of the neighborhoods of personalization. Finally, $\upsilon$ regularizes $\phi$ and should be set to reflect the user's prior knowledge about the influence of each covariate on personalization.

For our experiments, we use the following hyperparamters:

- *Simulation.* $\lambda = 1e{-}1$, $\gamma = 1e5$, $\upsilon = 1e{-}2$, $q = 2$
- *Finance.* $\lambda = 1$, $\gamma = 1e8$, $\upsilon = 1e{-}2$, $q = 50$
- *Cancer.* $\lambda = 1$, $\gamma = 1e6$, $\upsilon = 1e{-}2$, $q = 50$
- *Election.* $\lambda = 1e{-}2$, $\gamma = 1e3$, $\upsilon = 1e{-}2$, $q = 2$

For all experiments, we dynamically set $r$ such that each point has on average 10 neighbors, and use the learning rate schedule of $\alpha_0 = 1e{-}4$, $c = 1 - 1e{-}4$.

## B.4 Datasets

### B.4.1 Simulation

To estimate personalized models for the simulated dataset, we initialize the personalized estimations with a varying-coefficient model, and personalize according to the distance metric $d_1(x,y) = |x-y|$.

### B.4.2 Finance

The financial dataset is constructed by joining stock and ETF trading histories[1] to a database of global news headlines from Bloomberg [6] and Reddit[2]. We transform news headlines into continuous

representations by tf-idf weighting averaging [4] of word embeddings under the GLoVE model [26] pre-trained on Wikipedia and Gigaword corpora[3]. After dimensionality reduction, this news dataset consists of a 50-dimensional vector for each date. We split the dataset into training and test sets at the 80th percentile date, which is approximately the beginning of 2011. To estimate personalized models for the financial dataset, we intiialize the personalized estimators with the population model and personalize according to the $\ell_1$ distance for time and the discrete metric for the other covariates.

### B.4.3 Cancer

As described in the main text, we test the capacity of personalized models to distinguish benign from malignant skin cancers. The dataset contains 17,053 total samples from 17 patients. Each sample consists of 2,734 spectra intensities and is labeled with a binary outcome (0=benign, 1=malignant). Data from 9 patients are used to fit models, while data from 8 patients are held-out for evaluation. In this dataset, the only explicit covariate is the patient label. To produce covariates which are most useful for personalization, we augment the patient labels with 1500 of the predictive features compressed to 2 dimensions by t-SNE dimensionality reduction. These 1500 predictive features are excluded from the set of predictors for PR, while baseline methods use the entire set of features as predictors. We fit the personalized regression according to the distance function $d_1(x, y) = \mathcal{I}_{\{x \neq y\}}$, $d_2(x, y) = |x - y|$, $d_3(x, y) = |x - y|$, where the first function checks if the patients are the same and the final two calculate distance in the continuous covariates.

### B.4.4 Election

The election predictors are taken from the 2012 U.S. presidential election and consists of discrete representations of each candidate based on candidate positions compiled by ProCon.[4] Outcomes are the county-level vote proportions in the 2012 U.S. presidential election.[5] For the covariates $U$, we used county demographic information from the 2010 U.S. Census.[6]

## C  Additional figures and discussion

### C.1  Simulations

We adapt the procedure of Section 3.1 to higher $p$ by generating multidimensional $U$ and using a coordinate of $U$ to personalize each value in $\theta$. More precisely, we have $X \sim \text{Unif}(-1, 1)^p$, $U \sim \text{Unif}(0, 1)^K$, $a \sim \text{Unif}(0, 1)^p$, $b \sim \text{Unif}(0, 1)^p$, $c \sim \text{Cat}(K)^p$, $\theta_j = \mathcal{I}_{\{U_{c_j} > a_j\}} + b_j \sin U_{c_j}$, $Y^{(i)} = X^{(i)}\theta^{(i)} + N(0, 0.01)$. These experiments all use $K = 5$ covariates. PR outperforms baselines in all cases, and is strongest for large $n$, small $p$ (as expected).

### C.2  Finance

As described in the main text, we fit a variety of personalized models to a financial dataset of stock market price histories and world news headlines. Shown in Fig. S4 are visualizations of the model parameters, colored by each of the covariates used for personalization. We see that all of these covariates contribute to distribution of personalized models.

To understand the relevance of each feature, we visualize the coefficients for each security in Fig. S5. The striped pattern is a result of the alternating arrangement of news and prices. In all securities, the effects of the Great Financial Crisis in 2008 are clear. Interestingly, other recessions do not seem to have similar lasting effects on parameter values, implying that these recessions had fewer structural effects than the Great Financial Crisis.

### C.3  Cancer

Fig. S6 depicts the personalized regression models estimated for this task. We see that there is strong clustering according to patient label, indicating that patients have different "types" of tumor. However,

| $p$ | Model | $\|\hat{\Omega} - \Omega\|_2$ | $R^2$ | MSE |
|---|---|---|---|---|
| | Pop. | 9.97 | 0.87 | 0.13 |
| | MR | 9.86 | 0.88 | 0.12 |
| 2 | VC | 14.55 | 0.76 | 0.22 |
| | DNN | 30.42 | 0.75 | 0.24 |
| | PR | **7.82** | **0.89** | **0.09** |
| | Pop. | 15.19 | 0.79 | 0.73 |
| | MR | 14.81 | 0.80 | 0.70 |
| 10 | VC | 23.86 | 0.69 | 1.09 |
| | DNN | 67.49 | 0.80 | 0.85 |
| | PR | **14.52** | **0.82** | **0.65** |
| | Pop. | 25.86 | 0.85 | 1.26 |
| | MR | 25.75 | 0.86 | 1.20 |
| 25 | VC | 38.77 | 0.66 | 3.05 |
| | DNN | 103.72 | 0.68 | 2.78 |
| | PR | **24.53** | **0.87** | **1.10** |

Table S2: Simulations with $n = 500$.

| $n$ | Model | $\|\hat{\Omega} - \Omega\|_2$ | $R^2$ | MSE |
|---|---|---|---|---|
| | Pop. | 6.36 | 0.90 | 0.23 |
| | MR | 6.48 | 0.90 | 0.23 |
| 100 | VC | 10.75 | 0.78 | 0.50 |
| | DNN | 22.30 | 0.39 | 0.75 |
| | PR | **6.03** | **0.91** | **0.21** |
| | Pop. | 11.83 | 0.84 | 0.29 |
| | MR | 11.78 | 0.84 | 0.30 |
| 500 | VC | 19.06 | 0.74 | 0.49 |
| | DNN | 47.33 | 0.81 | 0.37 |
| | PR | **10.30** | **0.86** | **0.26** |
| | Pop. | 33.03 | 0.87 | 0.26 |
| | MR | 31.75 | 0.88 | 0.26 |
| 2500 | VC | 33.71 | 0.87 | 0.27 |
| | DNN | 102.88 | 0.88 | 0.29 |
| | PR | **26.11** | **0.90** | **0.21** |

Table S3: Simulations with $p = 5$.

this clustering by patient is not complete – there is also significant heterogeneity in the models for each patient. This may point to the view of "mosaic" tumors, in which multiple cell lines combine within single tumors [19].

Finally, by aggregating feature importance across each patient, we determine which feature is the most predictive of the sample labels for each patient (Table S4). We see that there are 9 different clusters, which are not accurately estimated by mixture models due to the sample heterogeneity within each mixture.

Table S4: Most predictive features, aggregated over each patient.

| Patient ID | Molecular Weight |
|---|---|
| 0 | 163.627 |
| 1 | 191.834 |
| 2 | 566.833 |
| 3 | 177.541 |
| 4 | 234.167 |
| 5 | 163.125 |
| 6 | 191.834 |
| 7 | 177.541 |
| 8 | 113.083 |
| 9 | 566.833 |
| 10 | 163.125 |
| 11 | 231.958 |
| 12 | 234.666 |
| 13 | 191.834 |
| 14 | 163.627 |
| 15 | 177.541 |
| 16 | 163.627 |

### C.4 Election

As described in the main text, we fit personalized models to a dataset of election results. Representations of the personalized models for Pennsylvania counties are shown in Fig. S8, with a key to the abbreviations in Table S5. These embeddings show that the demographics (Fig. S8a) do not completely correspond to voting outcome, so using these factors to understand election preferences leaves out significant latent factors. In contrast, the personalized models (Fig. S8c) form structure which trades off fidelity to demographic data with voting outcome. These trends are not captured by

(a) Industry

(b) Region

(c) Security

(d) Time

Figure S4: Representations of the model parameters fit to the financial dataset. Each point represents a single sample, colored according to covariates.

the baseline methods, such as the varying-coefficients model (Fig. S8d). In addition, concatenating the demographic and voting outcomes does not recover the same structure (Fig. S8e). These patterns are replicated in the election of 2008 (Fig. S9).

(a) AAPL

(b) AMZN

(c) BP

(d) BRK-B

(e) CHIX

(f) E

(g) FB

(h) GM

(i) GOOGL

(j) HSBC

(k) IEO

(l) JPM

(m) LMT

(n) MU

(o) NVDA

(p) OA  (q) RDS-A  (r) SNP

(s) SPY  (t) TSLA  (u) VOO

(v) WMT  (w) XIV  (x) YANG

Figure S5: Visualizations of the models fit to each security over time. The vertical axis indexes time, while the horizontal axis indexes features. Features are arranged according to each day of a two-week time span for each prediction, with alternating news and stock histories in each day.

Figure S6: Personalized models for patients in the training set of the cancer dataset. Each point represents a model for a single sample, colored by the patient ID. There is strong clustering according to patient label, but also intra-patient heterogeneity (notably Patients 1,3,4, and 6).

Figure S7: Histogram of the number of skin cancer samples for which each feature is the most predictive according to the magnitude of coefficients of the personalized models.

(a) Demographics, U

(b) Outcome, Y

(c) Personalized Estimation, $\widehat{Z}$

(d) VC Embeddings.

(e) Concatenated Embeddings.

Figure S8: Embeddings of Pennsylvania counties. Each point represents the t-SNE embedding of a representation of a county, with color gradient corresponding to the 2012 election result (red for Republican candidate, blue for Democratic candidate). (a) The county demographics (U) lie near a low-dimensional manifold that does not correspond to voter outcome. (b) The observed voting results lie near a one-dimensional manifold. (c) Personalized regression produces sample embeddings ($\widehat{Z}$) that interpolate between demographic and voting information.

(a) Demographics, U

(b) Voting Outcome, Y

(c) Personalized Estimation, $\widehat{Z}$

(d) Varying-Coefficients

(e) Concatenated

Figure S9: Embeddings of Pennsylvania counties. Each point represents the t-SNE embedding of a representation of a county, with color gradient corresponding to the 2008 election result (red for Republican candidate, blue for Democratic candidate).

Table S5: Abbrevations of Counties

| Abbreviation | Full Name |
| --- | --- |
| Alle. | Allegheny |
| Mont. | Montour |
| York | York |
| Faye. | Fayette |
| Adam. | Adams |
| Unio. | Union |
| Carb. | Carbon |
| Fore. | Forest |
| Perr. | Perry |
| Came. | Cameron |
| Pott. | Potter |
| Clin. | Clinton |
| Daup. | Dauphin |
| Merc. | Mercer |
| Fult. | Fulton |
| Cent. | Centre |
| Dela. | Delaware |
| Mont. | Montgomery |
| Warr. | Warren |
| Pike | Pike |
| Lehi. | Lehigh |
| Schu. | Schuylkill |
| Miff. | Mifflin |
| Susq. | Susquehanna |
| Juni. | Juniata |
| Bedf. | Bedford |
| Luze. | Luzerne |
| Brad. | Bradford |
| Lack. | Lackawanna |
| Some. | Somerset |
| Elk | Elk |
| Butl. | Butler |
| Erie | Erie |
| Lyco. | Lycoming |
| Sull. | Sullivan |
| Indi. | Indiana |
| Ches. | Chester |
| Monr. | Monroe |
| Nort. | Northampton |
| Craw. | Crawford |
| Arms. | Armstrong |
| Leba. | Lebanon |
| Cumb. | Cumberland |
| Camb. | Cambria |
| Hunt. | Huntingdon |
| West. | Westmoreland |
| Colu. | Columbia |
| Buck. | Bucks |
| Berk. | Berks |
| Clar. | Clarion |
| Vena. | Venango |
| Lanc. | Lancaster |
| Snyd. | Snyder |
| Fran. | Franklin |
| McKe. | McKean |
| Clea. | Clearfield |

## Footnotes

[1]https://www.kaggle.com/borismarjanovic/price-volume-data-for-all-us-stocks-etfs/version/3

[2]https://www.kaggle.com/aaron7sun/stocknews

[3]https://nlp.stanford.edu/projects/glove

[4]https://2012election.procon.org/view.source-summary-chart.php

[5]https://dataverse.harvard.edu/dataset.xhtml?persistentId=hdl:1902.1/21919

[6]https://www.census.gov/data/datasets/2016/demo/popest/counties-detail.html