[Reviews · NeurIPS 2019]

Reviewer 1



This paper presents a model for performing personalized regression; i.e. allowing for individual prediction models for each sample, rather than estimating a model for a group of samples. This is clearly useful in personalized medicine, but could also be applied in other settings, e.g. voter behaviour as shown in the paper. The method uses a linear model (although it could be extended to glms). Estimation is made feasible by reducing the p*n space of parameters to a lower-dimensional subspace via factor analysis. Additionally, regularization between parameter vectors for different samples is applied via a learned distance metric consisting of a weighted sum over covariate-specific base distances. Prediction for unseen datapoints is achieved via a nearest-neighbour approach using the learned distance function. The authors test the approach using a simulation study, as well as applying it to three real-world datasets from finance, medicine and voting records. The method seems original, and I am not aware of other approaches that share all of the characteristics of the one presented here. Regarding the quality, the method is well-motivated and robust, and the evaluation is thorough. In particular, I appreciate that the authors compared with a number of different methods, including a mixture regression model and a deep neural network. The simulation study could have been a bit more extensive; in particular, what is missing is an exploration of the effect of varying n and p. The work is well-presented and explained, and I had no issues following the details . However, there is one exception, which is that the subgradient approach requires (of course) the gradients of the objective function with respect to the parameters. These have not been included in either the main text or the supplementary material as far as I can tell. Perhaps they are easy to derive, but it would aid reproducibility to include these. In terms of significance, the improvement over the methods compared with in this paper is certainly impressive on some of the real-world datasets. Overall I would say that the method is presenting a number of incremental improvements over existing methods, that taken together amount to an important contribution to the literature. Edit after author response: The authors have provided a detailed response which addresses several of my concerns, including expressions for the subgradients and a more detailed simulation study. As a consequence, I have increased my score to an 8. Minor points: - For the prediction, it seems that only one nearest neighbour is used to predict the response for an unseen data point. This seems prone to error (what if the model for that nearest neighbour is not estimated well?). Could the authors discuss if there are ways to make the prediction more robust, e.g. by considering several nearest neighbours. - For the election dataset, the MR model seems to be doing exceedingly badly (much worse than a mean estimate) if the negative R^2 is to be believed. Why is this? - Reproducibility checklist: The authors say that they have included "an analysis of the complexity". I could not find this.

Reviewer 2



This paper presents a novel method to estimate mixture models by matching structure in sample covariates. It is clearly written and easy to understand. The proposed method is quite straightforward, and therefore the originality may not be strong enough. In addition, since the model is very complicated, it might suffer overfitting problems with noise data. Itt is also hard to scale and apply the proposed method to big data using popular cloud computing infrastructure today.

Reviewer 3



Overall the manuscript is well written and very accessible. The authors introduce a nice idea to personalize prediction models based on individual samples. They avoid over-fitting by constraining the matrix of personalized parameters to be low-rank. Furthermore, they introduce a regularization scheme as a second option, encouraging model parameters to be similar if the covariates are similar. The manuscript could benefit from drawing a connection to unsupervised domain adaptation methods, maybe at the point, where the distance matching approach is introduced.

[Author Response · NeurIPS 2019]

We thank the reviewers for their time and thoughts. Please find below responses specific to each reviewer's comments:

R1 • Please find more simulation results as well as a discussion of overfitting below.

• We thank the reviewer for noting that our specification of exact sub-gradients was buried in the Analysis section of
the supplement (L420-422); we will add a brief section detailing these sub-gradients used in our experiments.

• A brief discussion of computational complexity is contained in the "Scalability" Section of the main text. We can
expand this in the extra space provided on acceptance.

• The surprisingly poor quality of the MR predictions on the election dataset is due to overfitting – the MR outperforms
mean prediction on the training set, but these patterns do not transfer to the test set of other counties.

R2 • We'd like to emphasize that PR is *not a mixture model*. For example, we urge R2 to consider how a mixture model
would estimate parameter values for a component with only a single sample. This estimation would be undefined for
mixture models but is well-defined for PR. Thus, we consider the PR method qualitatively distinct from traditional
mixtures, and the paper should not be evaluated as simply adding a regularizer on traditional mixture models.

• Please refer to Section B.2 of the Supplement for a discussion of hyperparameter selection, as noted at L154–155.

• While we agree that scalability is an important problem, the clear focus of this paper is on modeling. Moreover,
although we not consider distributed or parallel implementations in this paper, the datasets we used can hardly be
described as "small": for instance, the finance dataset contains over 1000 features for over 14000 samples. We
certainly agree that parallelization of machine learning methods using tools like MapReduce and Spark improves
scalability, but this is not our focus. As an example, DNNs proved useful in applications well-before modern
distributed platforms for scaling DNN training (e.g. TensorFlow) were developed.

R3 • We thank the reviewer for pointing out similar motivations in recent unsupervised domain adaptation techniques,
which seek to adjust model parameters to the target domain. We are also interested to see further work expanding the
idea of using task representations (encoded via covariates in PR) to improve domain adaptation algorithms.

• Clarification: We emphasize that the low-rank and regularization schemes work together to jointly reduce overfitting
and to promote similarity. Our interpretation of this is that model constraints (via low-rank formulation) and
regularization (via DMR) must simultaneously be considered for sample-specific estimation to be successful.

• Please refer to Section C of the Supplement for a discussion of the benefits of PR on specific examples.

**Simulations.** Here we test the capacity of PR with varying $n$ and $p$. We adapt the procedure of Section 3.1 to higher $p$ by generating
multidimensional $U$ and using a coordinate of $U$ to personalize each value in $\theta$. More precisely, we have $X \sim \text{Unif}(-1, 1)^p$,
$U \sim \text{Unif}(0, 1)^K$, $a \sim \text{Unif}(0, 1)^p$, $b \sim \text{Unif}(0, 1)^p$, $c \sim \text{Cat}(K)^p$, $\theta_j = \mathcal{I}_{\{U_{c_j} > a_j\}} + b_j \sin U_{c_j}$, $Y^{(i)} = X^{(i)}\theta^{(i)} + N(0, 0.01)$.
These experiments all use $K = 5$ covariates. PR outperforms baselines in all cases, and is strongest for large $n$, small $p$ (as expected).

| $p$ | Model | $\|\|\hat{\Omega} - \Omega\|\|_2$ | $R^2$ | MSE |
|---|---|---|---|---|
| 2 | Pop. | 9.97 | 0.87 | 0.13 |
| | MR | 9.86 | 0.88 | 0.12 |
| | VC | 14.55 | 0.76 | 0.22 |
| | DNN | 30.42 | 0.75 | 0.24 |
| | PR | **7.82** | **0.89** | **0.09** |
| 10 | Pop. | 15.19 | 0.79 | 0.73 |
| | MR | 14.81 | 0.80 | 0.70 |
| | VC | 23.86 | 0.69 | 1.09 |
| | DNN | 67.49 | 0.80 | 0.85 |
| | PR | **14.52** | **0.82** | **0.65** |
| 25 | Pop. | 25.86 | 0.85 | 1.26 |
| | MR | 25.75 | 0.86 | 1.20 |
| | VC | 38.77 | 0.66 | 3.05 |
| | DNN | 103.72 | 0.68 | 2.78 |
| | PR | **24.53** | **0.87** | **1.10** |

Table 1: Simulations with $n = 500$.

| $n$ | Model | $\|\|\hat{\Omega} - \Omega\|\|_2$ | $R^2$ | MSE |
|---|---|---|---|---|
| 100 | Pop. | 6.36 | 0.90 | 0.23 |
| | MR | 6.48 | 0.90 | 0.23 |
| | VC | 10.75 | 0.78 | 0.50 |
| | DNN | 22.30 | 0.39 | 0.75 |
| | PR | **6.03** | **0.91** | **0.21** |
| 500 | Pop. | 11.83 | 0.84 | 0.29 |
| | MR | 11.78 | 0.84 | 0.30 |
| | VC | 19.06 | 0.74 | 0.49 |
| | DNN | 47.33 | 0.81 | 0.37 |
| | PR | **10.30** | **0.86** | **0.26** |
| 2500 | Pop. | 33.03 | 0.87 | 0.26 |
| | MR | 31.75 | 0.88 | 0.26 |
| | VC | 33.71 | 0.87 | 0.27 |
| | DNN | 102.88 | 0.88 | 0.29 |
| | PR | **26.11** | **0.90** | **0.21** |

Table 2: Simulations with $p = 5$.

**Noise and Overfitting.** We thank reviewers for asking about noise and overfitting. We see that Eq. 8 is incomplete; while it describes
a 1-NN procedure for selecting test models, we actually use $k$-NN by averaging the models of the $k$ nearest training samples. Our
experiments all use $k = 3$ neighbors. As we increase $k$, the test model approaches the population estimator (according to Eq. 7, the
barycenter of all sample-specific models has not moved far away from the population estimator). We will fix this description of Eq. 8
for the camera-ready version by including the averaging step. Finally, we believe the out-of-sample prediction results provide strong
evidence that any harmful overfitting of PR is outweighed by the benefit of personalized estimation. This agrees with famous results
such as [1], where it is showed that optimal ensembles of linear models consist of overfitted atoms; see also Eq. 12 and Fig. 2 therein.

[1]. P. Sollich and A. Krogh. Learning with ensembles: How overfitting can be useful. *Advances in Neural Information Processing*
*Systems*, pages 190-196, 1996.


[Meta-Review · NeurIPS 2019]

The paper introduces a new method for personalized regression, and tests it empirically on several problems. This is overall a nice contribution, the reviewers found that the paper brings a novel solution to a specific but common problem, and were overall happy with the detailed experimental study that supports the relevance of the proposed method on a variety of tasks. However, the novelty of the work is a bit limited given the extensive previous work on multitask learning and domain shift adaptation. The lack of theoretical analysis of the procedure (probably difficult due to the fact that the proposed penalty is not convex) also limits the scope of the contribution.